# Microbiota and Resistome Analysis of Colostrum and Milk from Dairy Cows Treated with and without Dry Cow Therapies

**DOI:** 10.3390/antibiotics12081315

**Published:** 2023-08-14

**Authors:** Dhrati V. Patangia, Ghjuvan Grimaud, Kevin Linehan, R. Paul Ross, Catherine Stanton

**Affiliations:** 1School of Microbiology, University College Cork, T12 K8AF Cork, Ireland; dhrati.patangia@teagasc.ie (D.V.P.); p.ross@ucc.ie (R.P.R.); 2Biosciences Building, Teagasc Food Research Centre, P61 C996 Fermoy, Ireland; 3APC Microbiome Ireland, University College Cork, T12 K8AF Cork, Ireland

**Keywords:** cow milk microbiota, resistome, shotgun metagenomics, longitudinal

## Abstract

This study investigated the longitudinal impact of methods for the drying off of cows with and without dry cow therapy (DCT) on the microbiota and resistome profile in colostrum and milk samples from cows. Three groups of healthy dairy cows (*n* = 24) with different antibiotic treatments during DCT were studied. Colostrum and milk samples from Month 0 (M0), 2 (M2), 4 (M4) and 6 (M6) were analysed using whole-genome shotgun-sequencing. The microbial diversity from antibiotic-treated groups was different and higher than that of the non-antibiotic group. This difference was more evident in milk compared to colostrum, with increasing diversity seen only in antibiotic-treated groups. The microbiome of antibiotic-treated groups clustered separately from the non-antibiotic group at M2-, M4- and M6 milk samples, showing the effect of antibiotic treatment on between-group (beta) diversity. The non-antibiotic group did not show a high relative abundance of mastitis-causing pathogens during early lactation and was more associated with genera such as *Psychrobacter*, *Serratia*, *Gordonibacter* and *Brevibacterium*. A high relative abundance of antibiotic resistance genes (ARGs) was observed in the milk of antibiotic-treated groups with the Cephaguard group showing a significantly high abundance of genes conferring resistance to cephalosporin, aminoglycoside and penam classes. The data support the use of non-antibiotic alternatives for drying off in cows.

## 1. Introduction

The first milk produced after the non-lactating period, colostrum, is highly nutrient-rich, provides the calf with both nutritional and immune benefits, and shapes the calf gut microbiota [1]. It helps in the education, maturation and development of the immune, organ system and intestinal function [2,3]. This is crucial for the calf as it is born agammaglobulinemic and depends on the colostrum for passive immunity and gut microbiota development [1]. Colostrum and stable gut microbiota colonisation are important in early life for ruminant development from the monogastric stage at birth, and impact the overall health of the animal [4]. Due to the importance of animal health and its relation to dairy products, research is now focused on bovine colostrum and milk microbiota composition and its association with disease [5,6,7,8].

In cows, before the milking begins, there is a period called the dry period (six to eight weeks long), which allows the teats to dry up. This can be performed by allowing the drying to occur naturally, using teat sealants. Alternatively, drying of the teats can be performed by using medications and antibiotics to ensure the teats are free of any infection, called dry cow therapy (DCT). The objectives of DCT are to eliminate infections presenting from previous lactation and to avoid new udder infections during the dry period—when mammary glands are highly susceptible to new infections [9]. The use of DCT can be either selective or for the entire herd. In the second approach, called the “blanket” approach, the entire herd is treated with antibiotics for preventive measures. It helps to achieve the goal of attaining high quantities of good-quality milk [9,10]; however, prophylactic treatment of animals with antibiotics is increasingly being frowned upon due to problems associated with antibiotic resistance development.

The use of antimicrobials for DCT is common globally [11,12,13,14], although a declining trend is now observed in many European countries [15]. Many antibiotics used in livestock are medically relevant and based on the same anti-microbial compound used in human medicine. Commonly used antibiotics include penicillin alone or in combination with aminoglycosides and cephalosporins, tetracyclines, fluoroquinolones, macrolides and sulfonamides [15,16].

Milk from cows treated with prophylactic antibiotics usually contains significant antibiotic residues [17,18,19]. Sub-MIC (minimum inhibitory concentration) levels of antibiotics can form resistant bacteria, which can be more problematic than those selected at higher doses [20,21]. An indirect role of colostrum as the diet is suggested as a potential source of antibiotic resistance genes (ARGs) for calf gut microbiota even in the absence of antibiotic use in cows [22]. Further, reports suggest the presence of ARGs and a significantly altered microbial and functional profile in the calf gut when fed with milk containing antibiotic residues [22,23,24,25,26]. Antibiotic resistance in dairy milk is also an area of interest because of the human consumption of raw milk or other dairy products made from contaminated raw milk and issues of environmental pollution with ARGs making it a One-Health issue [27,28,29,30,31,32]. Prophylactic use of antibiotics on dairy farms is potentially an important contributor to AMR spread in the environment. Thus, along with EU legislation to cease blanket DCT, other national laws to reduce antibiotic use in other livestock and dairy activities might help lower the development and spread of ARGs.

Moreover, the presence of ARGs is not limited to pathogenic bacteria but may also occur in commensals. Mostly, the antibiotic resistance of pathogens in bovine milk has been studied in cases of mastitis [33,34,35], but not many studies have been conducted to study the resistance pattern of the overall milk microbiota. Also, most studies focus on the resistance patterns of pathogens at one time point; thus, temporal effects of antibiotics on the resistance pattern in colostrum and milk are yet to be investigated. We hypothesised that antibiotic use in the Cephaguard^TM^ (CEF) (Cephaguard DC 150 mg intramammary ointment at drying off, of which the active ingredient is Cefquinome—a fourth-generation cephalosporin) and Ubro red (UBRO) (Ubro Red Dry Cow Intramammary Suspension at drying off; of which the active ingredients are Framycetin Sulphate, Penethamate Hydriodide and Procaine Penicillin) group during DCT will result in varying abundance and composition of microbiota and resistance genes as compared to the non-antibiotic group (NOAB). Thus, in this study, shotgun metagenomics sequencing was used to investigate the longitudinal microbial composition and resistome profiles of colostrum and milk samples obtained from healthy cows throughout lactation, which were previously treated with and without DCT.

## 2. Results

### 2.1. Microbiota Diversity Is Influenced by Antibiotic Treatment

Shotgun sequencing of the raw bovine colostrum and milk samples yielded 494,107,934 reads. Post trimming and host removal, the number of microbial reads remaining were 108,774,030 with an average of 1,121,382 reads (median = 310,854, min = 40,371, max = 6,941,451). Both Shannon and Chao1 diversity indices did not show a variation between time points within the no-antibiotic group (Figure 1A and Appendix A). However, significant differences were observed for both Shannon and Chao1 diversity indices between M0 and M2 (for UBRO: Shannon *p*-value = 0.026, Chao1 *p*-value = 0.011; for CEF: Shannon *p*-value = 0.00016, Chao1 *p*-value = 0.003; Wilcoxon test), M0 and M4 (for UBRO: Shannon *p*-value = 0.00058, Chao1 *p*-value = 0.0021; for CEF: Shannon *p*-value = 0.00031, Chao1 *p*-value = 0.0061; Wilcoxon test), and M0 and M6 (for UBRO: Shannon *p*-value = 0.00058, Chao1 *p*-value = 0.002; for CEF: Shannon *p*-value = 0.00016, Chao1 *p*-value = 0.0044; Wilcoxon test) in both Cephaguard (CEF)- and UBRO red (UBRO)-treated groups (Figure 1A and Appendix A). Additionally, a significant difference between M2 and M6 was observed for the Chao1 index in the UBRO group (Appendix A). Concerning individual time points, no significant differences were observed between the three groups at M0, while CEF and NOAB groups showed differences at M2; UBRO and NOAB at M4; and both antibiotics groups to NOAB at M6 (Figure 1B and Appendix A). The beta diversity of milk microbiota between the NOAB and antibiotic groups (UBRO and CEF) demonstrated distinct clustering overall and at all time-points after M0 (Figure 1C,D), including NOAB vs. CEF (*p* = 0.0030) and NOAB vs. UBRO (*p* = 0.0045) at M2, NOAB vs. CEF (*p* = 0.0015) and NOAB vs. UBRO (*p* = 0.0015) at M4, and NOAB vs. CEF (*p* = 0.0015) and NOAB vs. UBRO at M6 (*p* = 0.0015). Furthermore, discrete clustering of samples was observed based on time points, with samples at M0 clustering separately from all later time points (Figure 1D).

Similar to distinct clustering of the NOAB group from antibiotic-treated groups overall, we also observed clustering based on farms, where farm “C” (i.e., the farm without DCT) clusters separately from the other antibiotic-treated farms (farm “D” and “P” are CEF, while farm “L” and “T” are UBRO). Furthermore, samples from nulliparous cows, which corresponded to the NOAB group, showed a similar clustering pattern because of an overlapping between the antibiotics groups and parity (Appendix A–C, Appendix A). Based on PERMANOVA tests, the variable farms and antibiotics were the most explanatory variables regarding beta-diversity (Bray–Curtis distance) in our dataset (PERMANOVA R^2^ = 0.15 and R^2^ = 0.11, reciprocally). To test the interactions between antibiotics and farms, we performed a PERMANOVA on the two most explanatory grouping variables. The results showed that grouping based on antibiotic exposure explains a higher variance compared to farms (R^2^ = 0.11 and 0.03 and *p*-values = 0.001 and 0.02 for group and farm study variables, respectively; PERMANOVA using Bray–Curtis distance ~ Antibiotic groups + farm). These results considering the effect of farm variables must be interpreted cautiously due to the low subject size for each farm.

### 2.2. Taxonomic Composition Is Associated with Dry Cow Therapy Treatment

Differences in the relative abundance of the microbial composition of milk between the three groups at all time points were observed using plots at phyla and genera levels (Figure 2). Overall, the phylum Actinobacteria was observed to have high relative abundance in both colostrum and milk samples in all groups over time (Figure 2A). The next most abundant phylum was Proteobacteria, followed by Firmicutes and Bacteroidetes (Figure 2A).

The top 10 genera found in milk from each group over all time points were examined and several genera identified in cow milk in our study include *Acinetobacter*, *Brevibacterium*, *Corynebacterium*, *Lactobacillus*, *Lactococcus*, *Microbacterium*, *Pseudomonas*, *Propionibacterium*, *Kocuria* and *Staphylococcus* (Figure 2B). In the NOAB group, the genera *Actinoallotecihus*, *Corynebacterium*, *Brachybacterium* and *Microbacterium* were amongst the top 10 over all time points. *Brachybacterium* and *Brevibacterium* showed an increase in the no-antibiotic group, compared with the antibiotic-treated groups. Further, *Corynebacterium* was found in both antibiotic-administered and NOAB groups with higher initial abundance in the NOAB group. *Actinoalloteichus* was also present in all three groups in high relative abundance at M0 and showed a declining trend with time of lactation (Figure 2C). A higher abundance of *Acinetobacter* was observed in the antibiotic-administered groups compared to NOAB. Additionally, *Rhodococcus*, *Ottowia* and *Lactobacillus* were observed in higher relative abundance in both the antibiotic-treated groups and demonstrated an increasing trend over time of lactation. *Pseudomonas*, *Microbacterium*, *Corynebacterium* and *Kocuria* were also present at all time points in all three groups, with *Kocuria* and *Microbacterium* increasing over lactation time in the antibiotic-treated groups.

A closer look at the relative abundance of major mastitis-causing pathogens in dairy cows revealed a lack of high relative abundance of potential pathogens such as *Escherichia* and *Staphylococcus* in the non-antibiotic group at M0 and M2 (Figure 3A). The relative abundance of *Streptococcus* was close to zero in all three groups (plot not shown). The relative abundance of *Corynebacterium* was higher in the NOAB group than the CEF group at M0, and higher than both antibiotic-treated groups at M2 (Figure 3A).

Relative associations of genera to each group were estimated with Songbird [36] using the formula: C(Group, Treatment(‘NOAB’)) (Figure 3B). Upon comparing the antibiotic-treated group with the no-antibiotic group, *Brevibacterium*, *Brachybacterium*, *Serratia* and *Gordonibacter* were more associated with the NOAB group, while *Ottowia*, *Kocuria*, *Rhododcoccus*, *Microbacterium* and *Pseudomonas* were more associated with the antibiotic-treated groups. A comparison of the differentially abundant genera between the antibiotic-treated groups (UBRO and CEF) did not show any pattern, highlighting a stronger association of any genera to a particular antibiotic group.

### 2.3. Impact of Dry Cow Therapy on Antibiotic Resistance Gene Reservoir

RGI-bwt was used to predict antibiotic resistance genes in filtered and quality-controlled shotgun sequencing reads. ARGs belonging to 189 AMR gene families conferring resistance to 43 different classes of antibiotics were found. Of these, several genes conferred resistance to more than one class and genes conferring resistance to three or more classes were termed multi-drug-resistance (MDR) genes (Figure 4A). Further, resistance was also observed for classes not administered in the present cycle of DCT with a very high abundance of MDR genes (Figure 4B).

The alpha diversity of ARGs over time was not different in milk samples from all groups (Shannon index), while the CEF and NOAB groups showed significant differences (Chao1 index), pointing towards higher richness in the CEF group. No significant differences were observed between groups or between time points in individual groups (Figure 5A). Beta diversity analysis showed a discrete clustering of milk microbiota between the CEF and NOAB groups (p.adj = 0.03, pairwiseAdonis) (Figure 5B). No significant clustering was observed at any individual time points except M6 (p.adj values: NOAB vs. CEF: 0.039, NOAB vs. UBRO: 0.015, pairwiseAdonis) (Figure 5C).

The CEF group showed the highest ARG abundance, with both antibiotic-treated groups showing a higher ARG abundance compared to the NOAB group (Appendix A). The abundance of genes conferring resistance to the drug class cephalosporin at M0 was higher in abundance in the CEF group. Multiple comparisons between groups (NOAB group as control) were performed for each class and high resistance was found in the antibiotic groups (Figure 6).

## 3. Discussion

In this study, we compared the microbial and resistome profiles of colostrum and milk from healthy cows throughout lactation, separated into three groups treated with differing dry cow therapies—two antibiotic-treated groups and one non-antibiotic-treated group using shotgun metagenomics. Studying the bovine milk microbiome is considered challenging as it is a low-biomass sample, difficult to collect without contamination, along with the possibility of kit contaminants, with an inter-individual variation in microbiota [6], different farming practices [37] and a lack of studies to develop a standardised protocol. Shotgun metagenomics is rarely used to study the resistance pattern of the microbiota in dairy colostrum and milk; however, this approach provides high resolution and allows functional profiling [38]. The antibiotics used were Cephaguard (cefquinome—fourth-generation cephalosporin) and Ubro red (active ingredient Framycetin Sulphate, Penethamate Hydroiodide and Procaine Penicillin), which are broad-spectrum antibiotics. Their target species include major mastitis-causing organisms such as *Streptococcus* spp. (*uberis*, *dysgalactiae*, *agalactiae*) and *Staphylococcus aureus*, which are coagulase-negative staphylococci for Cephaguard, while UBRO red targets *Staphylococcus* spp., *Streptococcus* spp., *Corynebacterium* spp., *Escherichia* spp., *Klebsiella* and *Pseudomonas* spp. The European Medicine Agency (EMA) classifies cefquinome as a category B (restricted use) antibiotic, while Framycetin is to be used with caution [39].

This study demonstrated significant differences in the diversity of microbial genera within the antibiotic-treated groups over the time points (Shannon diversity) similar to [40]. Similar results of increasing microbial diversity were also observed by Hermansson et al., (2019) [41] in breast milk of mothers treated with antibiotics, though the reason for it was unknown. However, the transition from colostrum to milk in the non-antibiotic group did not demonstrate significant changes in microbial diversity over time. This suggests that low initial microbial diversity in the antibiotic-groups could be due to antibiotic use. Further, the diversity between groups was significantly different at M2, M4 and M6 (beta diversity), with the NOAB group clustering distinctly compared to antibiotic-treated groups. Additionally, we do not report a high prevalence of mastitis-causing pathogens such as *Escherichia* spp., *Streptococcus* spp. and *Staphylococcus* spp. in the groups, especially in the NOAB group. Similar to other studies [40,42,43,44], our data support the use of only teat sealants for drying off, reducing antibiotic use.

The top 10 genera observed in our study were similar to those reported by earlier studies and include *Acinetobacter, Brevibacterium, Corynebacterium, Lactobacillus, Lactococcus, Microbacterium, Pseudomonas, Propionibacterium, Kocuria* and *Staphylococcus* (Figure 2B) [32,43,45,46,47,48,49,50]. *Acinetobacter* was high in antibiotic-treated groups while *Corynebacterium* was higher in the no-antibiotic group. The presence of *Corynebacterium* in the top 10 throughout lactation in the NOAB group is not unusual as it is thought to be controlled by various anti-sepsis measures [51,52]. Its high prevalence could also be because it is observed to be a dominant bacterium on teat skin [53]. Some studies suggest an association between *Corynebacterium* and intramammary infections, suggesting its role as a mastitis-causing pathogen, though this association is mainly with *Corynebacterium bovis*, while most other species are reported to be present in environmental niches [54]. Others suggest that the high abundance of *Corynebacterium* may provide protective effects against infections caused by major mammary gland pathogens such as *Staphylococcus* and *Streptococcus* [55,56]. To confirm the role of the high relative abundance of *Corynebacterium* in the NOAB group, further studies including a mix of culture and sequencing-based methods are needed. *Actinoalloteichus* was observed to have high relative abundance in all groups in the study. *Actinoalloteichus* was found in milk aliquots in a recent study [46], but not many other studies have reported its presence. *Actinoalloteichus* is reported to be isolated from marine sponges [57], seashores [58] and soil [59] and is considered to be a source of secondary metabolites due to the presence of several secondary metabolite biosynthetic gene clusters [60]. The high relative abundance of this genus in all three groups is surprising, and further studies are needed on Irish farms to study its origins.

The microbial diversity in milk from cows is affected by several factors. For instance, [45] demonstrated changes in microbial diversity longitudinally. The authors observed an increased abundance of *Streptococcus* and *Kocuria* over the summer months. The antibiotic-treated groups in our study show an increasing trend of *Kocuria* over lactation time, and these time points overlap with summertime in the given geographical area. Similarly, parity of the cow [61] and seasonal housing (indoor or outdoor sampling and housing) [62] can impact the milk microbial composition. Stage of lactation, milking hygiene, bedding material and feeding habits affect cow milk microbial composition and diversity [37,48,53,63]. For instance, Doyle et al. (2017) [62] observed that *Acinetobacter* and *Pseudomonas* were in lower proportions in milk from indoor-housed animals as compared to outdoor milk samples. An increase in these genera over M2 and M4 was observed in this study, which could be attributed to the fact that once the cows calve and are put on grass, they are outdoors full-time (later time points—M4 and M6—in our study correspond to outdoor samples). However, *Acinetobacter* and *Pseudomonas* are also psychrotrophic and observed in stored bulk milk samples [47] and are common dairy environment contaminants; thus, further studies are needed to investigate their origin in raw quarter milk samples. Another possible explanation as described by Oikonomou et al. (2020) [5] is that the milk microbiota was not alive and DNA from dead bacteria was thus not affected by treatments.

As reported earlier, we observed the presence of ARGs in cow colostrum; however, changes in ARG diversity over time were not observed in any group in this study. There are certain antibiotic resistance genes (ARGs) found in the NOAB group, while other ARGs that do not belong to commonly used antibiotic classes might be attributed to past antibiotic exposure or environmental factors such as contaminated water or manure, as well as horizontal gene transfer within the herd. Further, a high abundance of ARGs in the antibiotic-treated groups was observed. Certain genes conferring resistance to classes like beta-lactams, penams and cephalosporin and MDR genes were observed. The presence of ARGs (those that confer resistance to the antibiotic in question) in the antibiotic-treated groups can be justified due to the use of antibiotics, while many other resistance genes conferring resistance to drug classes like tetracycline, intercalating dye, disinfecting agents, triclosan and glycopeptide antibiotics could be due to contaminating bacteria from environmental sources such as irrigation [64], groundwater [65,66], slurry waste [67], manure and interaction with other animals in the herd [68,69]. This high ARG abundance detected in the antibiotic-treated groups could also be a result of the selection of MDR bacteria possessing multiple ARGs post-antibiotic use. Certain resistance could be due to previous antibiotic use and might be horizontally transferred between bacteria over time. The presence of ARGs in colostrum could result in the transfer to calves, leading to the further spread and dissemination of ARGs in the environment. Thus, this study lends further support to reducing the use of antimicrobials in dairy cows as well as their cautionary use in other farm practices, to reduce the development and spread of antibiotic-resistant bacteria.

The limitations of this study include the relatively small number of animals included per group, although we would emphasise that these animals were all followed longitudinally for six months of lactation. Another limitation of the study was the long sample intervals, which spans various seasonal and feeding variations; however, all cows belonged to the same geographic location; thus, the effect of these variables, if any, was similar throughout the dataset. We suggest that this study demonstrates the value of conducting temporal microbial and resistome studies on bovine colostrum and milk microbiota to examine the effect of DCT and antibiotic use. More studies with larger cohort sizes and shorter between-sample intervals are needed to understand the longitudinal effect of antibiotics on bovine colostrum and milk samples. This will help reduce antibiotic use in livestock and decrease the generation and transfer of ARGs.

## 4. Methods

### 4.1. Enrolment Criteria and Treatment for Cows

Colostrum and subsequent milk samples at month 2 (M2), month 4 (M4) and month 6 (M6) were collected from healthy cows from five different Irish farms between February and September 2020 as detailed below. All cows were housed in different herds in farms located in County Cork, Ireland. The farms included in our study were labelled farm D, farm P, farm L, farm T and farm C. Cows from each farm were selected randomly and divided into three groups based on the DCT treatment. Cows on all farms were treated with a teat sealant—Boviseal—in addition to antibiotics as described during the drying-off period. NOAB did not use any antibiotics during the drying-off period (natural drying off using teat sealant) and was assigned as the control group. The UBRO group was treated with blanket DCT using Ubro Red Dry Cow Intramammary Suspension at drying off. The CEF group was treated with blanket DCT using Cephaguard DC 150 mg intramammary ointment at drying off. Twenty-four cows divided into three groups were included in the study (NOAB: *n* = 9, CEF: *n* = 8, UBRO: *n* = 7), with four samples collected longitudinally from each cow throughout the first six months of lactation. In terms of the farm-wise distribution of the cows included in our study, all NOAB cows were from a single farm, Farm C (*n* = 9), while farm D and P constituted the CEF group and had *n* = 4 and *n* = 4, respectively. Farm L and T formed the UBRO group and had *n* = 5 and *n* = 2, respectively. Farms D and P reported routinely using Cephaguard while farms L and T routinely used Ubro red therapy. Cows showing signs of mastitis or any other infections were excluded from the study and milk samples were collected only from healthy cows. Cows were not subjected to antibiotic treatment during the time interval of sample collection.

### 4.2. Sample and Data Collection

All farms and personnel who collected samples included in this study were quality-assurance-certified members of the Irish Food Board—Bord Bia. Standard recommendations from the National Mastitis Council’s Laboratory Handbook on bovine mastitis were followed for sample collection [70]. Instructions to farmers were provided for sterile sample collection. Briefly, teats were disinfected with iodine tincture, dried and thoroughly disinfected with wipes soaked in 70% alcohol. Initial milk streams were discarded and then 15 mL of colostrum within the first hour post partum and milk samples from the front right quarter of each cow’s udder were collected into sterile 15 mL falcon tubes (Sarstedt) without preservatives. Colostrum (M0), and milk samples at month 2 (M2), month 4 (M4) and month 6 (M6) were collected from each cow. The samples were immediately frozen at −20 °C in chest freezers on the farms. Within a week of sample collection, the samples were transported to the laboratory where they were stored at −80 °C until further use. DNA extractions were performed within six months of sample collection. Metadata regarding the parity of cows, any previous infections, the treatment used for DCT and any other medications were collected from the farms (Appendix A).

### 4.3. DNA Extraction

Milk samples were thawed by placing them at 4 °C the night prior to DNA extractions. The samples were homogenised by inverting the tubes and centrifuged at 4000× *g* for 30 min at 4 °C. The fatty/cream layer that accumulated at the top was discarded using sterile cotton swabs and the supernatant was discarded. DNA was extracted from the pellet according to the manufacturer’s instructions using the PowerFood microbial DNA Isolation Kit (MoBIO Laboratories Inc., Carlsbad, CA, USA) with modifications as follows: Briefly, the cell pellets were washed with PBS (phosphate-buffered saline) (Sigma Aldrich, St. Louis, MO, USA) and centrifuged at 13,000× *g* for one minute at 20 °C to discard the supernatants. This process was repeated until the supernatant was no longer cloudy. The pellet was then re-suspended in PBS and treated with 90 µL of 50 mg/mL lysozyme (Sigma Aldrich, Lysozyme activity: ≥40,000 units/mg protein) and 50 µL of 5 KU/mL mutanolysin (Sigma Aldrich) followed by 15 min incubation at 55 °C with vortexing at intervals of 5 min. Further, samples were treated with Proteinase k (Qiagen, UK, 28 µL of 20 mg/mL, >600 mAU/mL), incubated at 55 °C for 15 min and then treated according to the manufacturer’s instructions using the PowerFood microbial DNA Isolation Kit protocol. The extracted DNA was stored at −30 °C until further use.

### 4.4. Amplification of DNA, Library Preparation and Shotgun Metagenomics Sequencing

The quantification of DNA was performed using the Qubit double-stranded DNA (dsDNA) high-sensitivity assay kit (Invitrogen). Some samples had less than 0.2 ng/µL of DNA; thus, a modified low-input Nextera XP protocol was used for shotgun sequencing library preparation as described previously [71]. Briefly, 5 µL of DNA was used for the tagmentation run along with 10 µL of Tagment DNA (TD) buffer and 5 µL of 1:10 diluted Amplicon Tagment Mix (ATM), making the total reaction volume 20 µL. The samples were incubated at 55 °C for 5 min on the thermal cycler for tagmentation reaction, with the immediate addition of 5 µL of NT (Neutralise Tagment) buffer to stop the tagmentation. Subsequent amplification was performed using 20 amplification cycles to ensure high quantities of amplified product for downstream reactions. A 1.6× ratio of Ampure XP beads was used to purify the libraries and the purified product was eluted in 20 µL of re-suspension buffer. The quality of purified DNA was determined using the High-Sensitivity DNA kit on the Agilent Bioanalyzer. The amplified DNA was quantified using Qubit and the samples were then pooled and outsourced for sequencing to the Teagasc Next-Generation DNA Sequencing Facility (Teagasc, Ireland).

### 4.5. Bioinformatics and Statistical Analysis

Raw metagenomics shotgun reads received post sequencing were quality-checked using FastQC (0.11.8) and MultiQC (v1.9). The reads were then filtered and trimmed using *Trim galore* (v0.6.1) and Bowtie2 (v 2.4.4) using the *Bos taurus* reference genome database from NCBI (Genome assembly ARS-UCD1.3) to remove any host contaminant reads. Reads obtained post-host removal step were considered microbial and were used for further analysis. The filtered and trimmed reads were then used for taxonomic assignment with Kraken2 (v2.1.1) and kraken-biom (https://github.com/smdabdoub/kraken-biom). Further, antibiotic resistance analysis was performed using RGI (Resistance Gene Identifier) (RGI-bwt) (v3.1.4) and CARD (Comprehensive Antibiotic Resistance Database) [72]. The RGI results were normalised to copies per million based on read counts using all mapped reads from the gene mapping output file of RGI-bwt. The results were further log-transformed and the abundance of ARGs at each time point was visualised using the heatmap, boxplots and bar plot in R. Songbird [36], using the formula C (Group, Treatment (‘NOAB’)), was used to generate differentials that described the association of taxa to the study groups. Songbird provides a comparatively greater ability to detect associations in compositional data analysis methods [73]. The differentials obtained (Appendix A) were ordered by rank and the top 10 most positive and negative genera were plotted and visualised using R (Figure 3B). Negative coefficient values denote stronger associations of the feature (genera in our analysis) with the reference. The phyloseq object [74] formed from Kraken2 output was normalised and filtered to keep reads with an abundance greater than 5 × 10^−4^ reads per taxa, and those belonging to viruses, Euryarchaeota and Archaea, and not classified at phylum level were removed. Initially, 5194 taxa were obtained, which were filtered down to 235 taxa. Stringent filtering parameters were applied to make sure that no taxa arising as contaminants/singletons remain in downstream analysis. Further, decontam package (v1.14.0) in R was used to look for contaminants in samples and remove them. Using the default parameters, no PCR blank reads were observed in any of the study samples. The negative control sample (with less than 500 reads) was thus excluded from further analysis and the phyloseq object with all study samples was used for further analysis. Downstream analysis was performed in R (v4.0.2) using packages phyloseq (v1.38.0), microbiome (v1.16.0) and ggplot2 (v3.3.2). Shannon and Chao1 diversity indices were used to calculate the alpha diversity of the microbiota within the groups. The Shannon diversity index provides details about richness and evenness while the Chao1 index estimates total richness only. The Wilcoxon test was performed to test if the difference between any two groups or time points was significant. Beta diversity analysis was performed using the Bray–Curtis dissimilarity matrix to determine compositional dissimilarities between the samples based on group, time, farm and parity variables in the study. The Bray–Curtis distance matrix takes into account abundance/occurrence data and does not rely solely on the presence or absence of data. The impact of various study variables on the microbiota diversity was examined using permutational multivariate analysis of variance (PERMANOVA). Statistical analysis was also performed in R using the Wilcoxon test and Dunn test with the function stat_compare_means and stat_pwc from package ggpubr (v0.4.0) and Permanova using the pairwiseAdonis (v0.4) package. *p*-values were adjusted using the false discovery rate (fdr) method in R unless specified otherwise.

## 5. Conclusions

Antibiotic treatment during DCT was associated with high abundances of ARGs in milk compared to the non-antibiotic control group, and altered microbial composition throughout lactation. Using teat sealant without antibiotics did not lead to the presence of a high abundance of mastitis-causing pathogens. Furthermore, the microbial diversity (alpha diversity) in antibiotic-treated groups increased over time, which was not observed in the non-antibiotic group. The reason for this is unknown; however, it could be due to effect of antibiotics on highly abundant bacteria, leaving a higher diversity of unaffected species. The presence of ARGs at high abundance in cow colostrum and milk is alarming because of the threat of passage of ARGs to the calf and to humans through the food chain. Both previous and current antibiotic use may result in the selection of bacteria that carry resistance genes, thus increasing the overall resistome profile diversity of the microbiota. It is challenging to trace the source of bacteria and thus the resistance genes in question; thus, preventive and protective approaches to reduce the use of antibiotics on farms and associated resources is crucial.

## Figures and Tables

**Figure 1 antibiotics-12-01315-f001:**
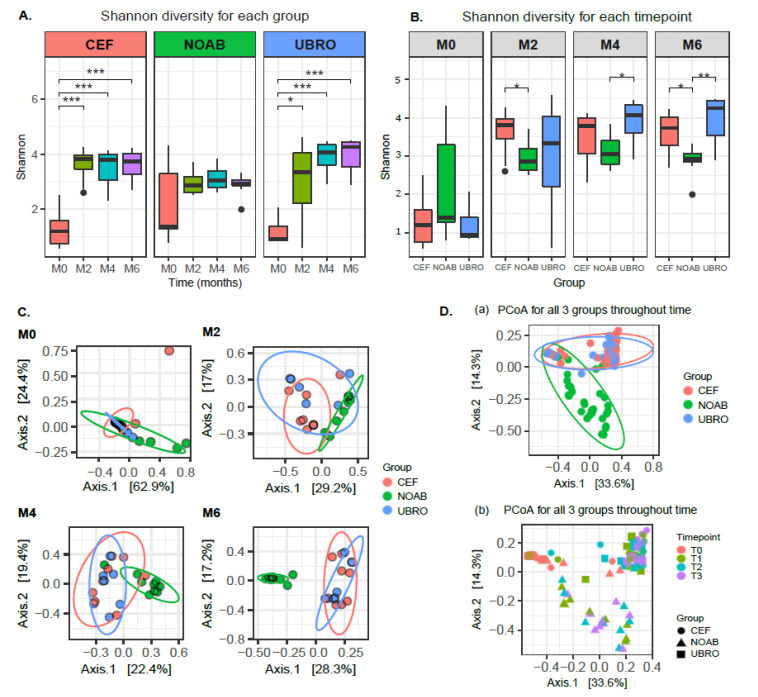
(**A**) Alpha diversity of milk microbiota using Shannon index between all time-points for each group. The plot shows significant differences in diversity between milk obtained at M0, M2, M4 and M6 for both the antibiotic-treated groups (UBRO and CEF). (**B**) Alpha diversity using Shannon index between the three groups (NOAB, UBRO and CEF) at all time-points. Plot shows no significant difference between groups at M0, while significance between antibiotic and no-antibiotic group was seen at later time points. (**C**) PCoA plot using Bray–Curtis distance matrix between the three groups at all time-points, showing distinct clustering between groups. (**D**) PCoA plot using Bray–Curtis distances (**a**) showing distinct clustering of all three groups, with antibiotic groups clustering discretely compared to the no-antibiotic group; and plot (**b**) showing separate grouping of points between groups and time-points. * *p*-value ≤ 0.05; ** *p*-value ≤ 0.01; *** *p*-value ≤ 0.001.

**Figure 2 antibiotics-12-01315-f002:**
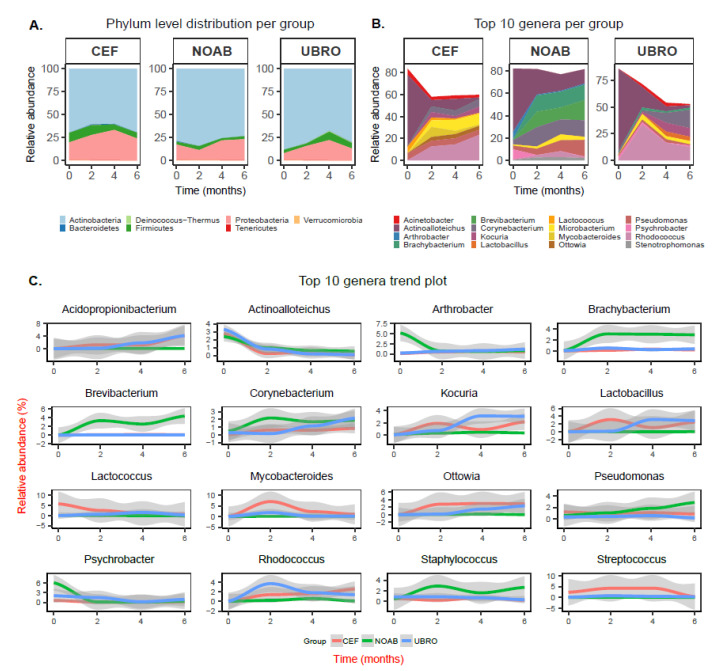
(**A**) Phylum level distribution of taxa in all three groups across all time points. (**B**) Overall top 10 genera in all three groups (NOAB, UBRO and CEF) at all time-points. (**C**) Trend plots at genera level, showing relative abundance of top 10 genera from all three groups (NOAB, UBRO and CEF) over time with confidence interval of 95% (with different *Y*-axis scale for each taxa).

**Figure 3 antibiotics-12-01315-f003:**
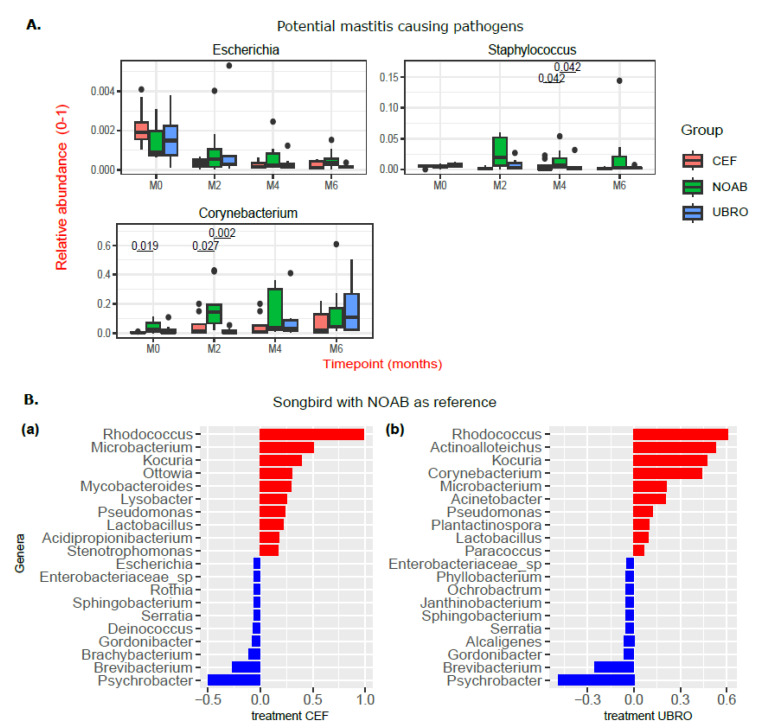
(**A**) Boxplots showing abundance of potential mastitis-causing pathogens in milk of all three groups (NOAB, UBRO and CEF) over time of lactation. Significance was determined using Wilcoxon test in R, and p.adj values < 0.05 are considered significant. The black dots correspond to the outliers. (**B**) Songbird differentials obtained using the formula C (Group, Treatment(‘NOAB’)) were sorted by ranks and the top 10 positive and negative features are plotted and depicted using bar charts. NOAB group is reference (negative here is more associated with reference group—NOAB here) and (**a**) and (**b**) are top 10 positive and negative associations with treatment group CEF and UBRO respectively.

**Figure 4 antibiotics-12-01315-f004:**
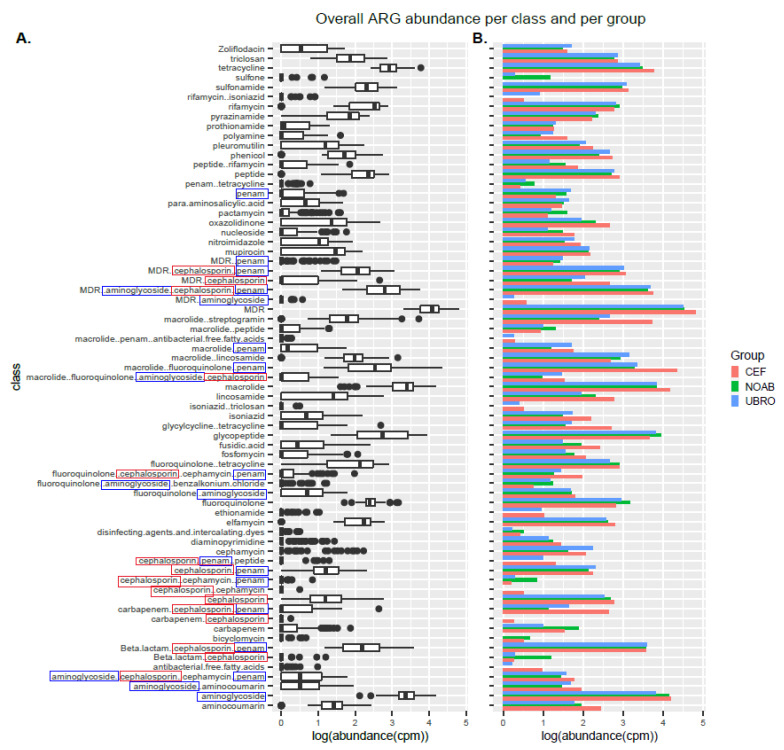
(**A**) Boxplot showing log-transformed abundance of the different classes of antibiotics to which resistance was observed in this study. (**B**) Bar plot showing representation of ARGs in all three groups (NOAB, UBRO and CEF) over all time-points. The classes highlighted in blue boxes correspond to the class of antibiotics administered to the UBRO group, while those in red boxes correspond to the CEF group.

**Figure 5 antibiotics-12-01315-f005:**
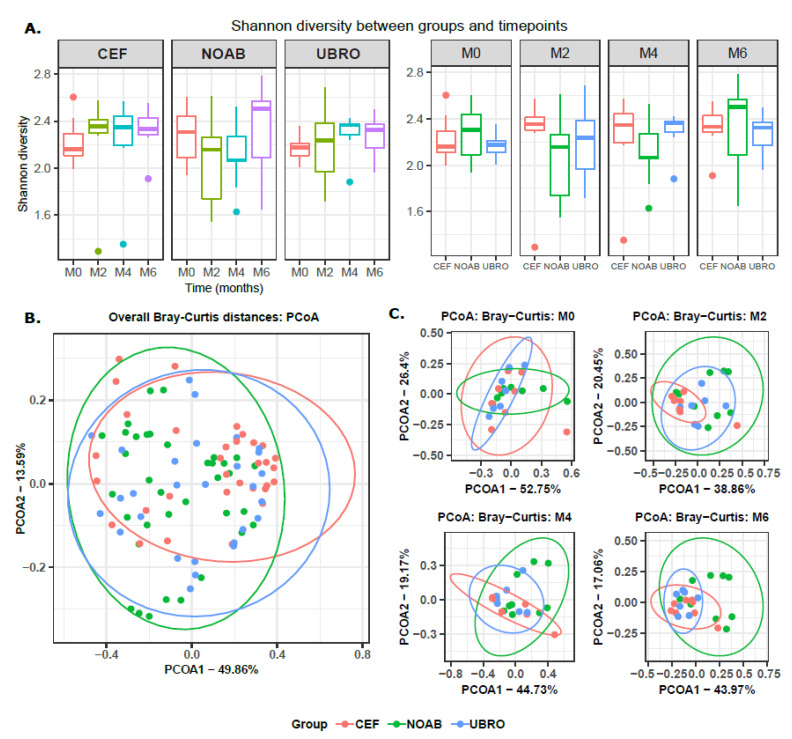
(**A**) Alpha diversity with Shannon index shows no difference between groups (NOAB, UBRO and CEF) at any time points for ARG abundance. (**B**) Beta diversity of ARGs using PCoA and Bray–Curtis distance. (**C**) Beta diversity of ARGs at each time point using PCoA plot with Bray–Curtis distance.

**Figure 6 antibiotics-12-01315-f006:**
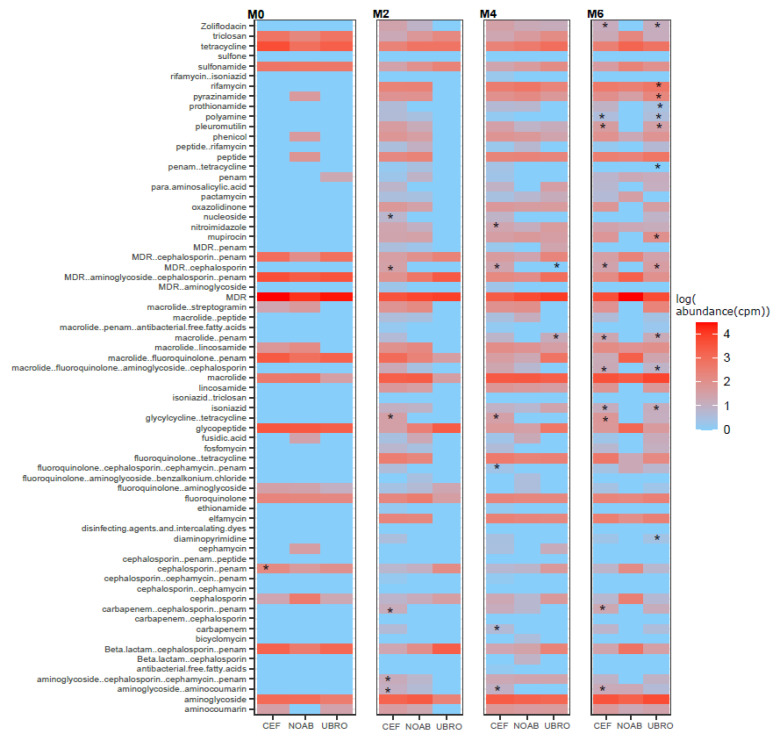
Heatmap showing log-transformed abundance (cpm) of various ARG classes per group at each timepoint. Significance was calculated with NOAB group as reference group against each antibiotic-treated group using Dunn test for multiple comparisons. P.adj values below 0.05 were considered significant and denoted with *.

## Data Availability

The data generated by shotgun sequencing as a part of this study were submitted to NCBI and are available under the project number PRJNA945243.

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
