# Peer review of "Microbiota and Resistome Analysis of Colostrum and Milk from Dairy Cows Treated with and without Dry Cow Therapies"

_antibiotics, 2023, doi:10.3390/antibiotics12081315_

Round 1

Reviewer 1 Report

The manuscript is edited at a good level and I have a few comments about the text:

Abstract

The limitations of this study include the relatively small number of animals - only 24 cows divided into 3 groups. I don't understand why the authors didn't create bigger groups when they had 5 farms at their disposal.

Explain the abbreviation ARG in the abstract.

Introduction

I recommend replacing the old citation Quigley et al., 2013 with a new one: https://doi.org/10.24425/pjvs.2022.143540

I recommend replacing the old citation Erskine et al., 2002 with a new one: https://doi.org/10.3390/ani12040470

Methods

On the basis of which parameters were cows selected from individual farms?

Is not clear number of cows included in the study. It was 24 cows from each farm or all farms?

Results

Statistical expression with commentary are at a very good level.

Discussion

More to describe the part about the presence of ARGs in cow colostrum.

Conclusion

The conclusion is clearly defined.

Reviewer 2 Report

In the paper, the author pointed microbiota and resistome analysis of colostrum and milk from dairy cows treated with antibiotic based dry cow therapies  I strongly recommend this article is worthy to publish in the journal after correction of some points described bellow:

1.       Why the use of preservatives was abandoned during sampling.

2.       Page 2; the abbreviations M2, M4, M6 should be explained

Author Response

Reviewer 2:

In the paper, the author pointed microbiota and resistome analysis of colostrum and milk from dairy cows treated with antibiotic based dry cow therapies  I strongly recommend this article is worthy to publish in the journal after correction of some points described bellow:

  1. Why the use of preservatives was abandoned during sampling.

We thank the reviewer for the comments and appreciate their constructive criticism and concerns, with a view to improving the quality of the current manuscript. We opted to use falcons without preservatives as we aimed to use the samples at the earliest possible (all DNA extractions were performed within 6 months of collection) and did not want any preservatives to interfere with our analysis.

  1. Page 2; the abbreviations M2, M4, M6 should be explained

We have now explained the abbreviations M2 (Month 2), M4 (Month 4) and M6 (Month 6) at page 2 and modified the manuscript.

Reviewer 3 Report

The manuscript entitled “Microbiota and resistome analysis of colostrum and milk from dairy cows treated with antibiotic-based dry cow therapies” aimed to determine the effect of dry cow therapy (DCT) including 2 groups with and 1 group without antibiotics on the microbiota and resistome profile in colostrum and milk samples from cows. Colostrum and milk samples from Months 0 (M0), 2 (M2), 4 (M4), and 6 (M6) were analyzed using whole-genome shotgun sequencing. Results show that microbial diversity from antibiotic-treated groups was significantly higher than the non-antibiotic group. The non-antibiotic group did not show higher numbers of mastitis-causing pathogens during early lactation. A higher antibiotic resistance gene was observed in the milk of an antibiotic-treated group. The result is interesting to discourage the use of blanket DCT in dairy farms in European countries.

In general, DCT is referred to as intramammary infusion of long-acting antimicrobial agents at drying off, so changing the word “DCT without antibiotics” to “without DCT” for the entire manuscript. Therefore, please correct the word DCT for the whole manuscript.

The major concern about the design study is that cows in groups (NOAB, CEF, and UBRO) were in different farms meaning that the diversity of microbiota and ARGs might be caused by the farm management or having selection bias data from farm or parity. The authors also demonstrated farm and cow parity effects on Page 5. The finding that mastitis pathogens were not different among groups might be caused by the difference in farm management. Please clarify that those diversity results were mainly affected by DCT, not by farm effects. The description in the result part is too long without any specific or important message given to readers. Please shorten and emphasize the important finding of the study. Too much information in the results caused the lack of specification, especially on the effects of times during lactation. Discussion should be emphasized on the important finding of the study.

Methods to describe data and analyze data should be in the materials and methods parts, not in the result part.

Please make clear this sentence “This can be done either by allowing the drying to occur naturally, using teat sealants, or by using medications and antibiotics to ensure the teats are free of any infection, called dry cow therapy (DCT).” that DCT is only the using long-acting antibiotic intramammary infusion. And check on that for the entire manuscript.

Page 2: Please write the full name for the first abbreviated use in the sentence “Colostrum is suggested as a potential source of ARGs for calf gut microbiota in the absence of antibiotic use in calves”. This sentence is in contrast, please check.

Page 2: The sentence “We hypothesized that antibiotic use in the CephaguardTM (CEF) and Ubro red (UBRO) group” should identify their active ingredients and source of the antibiotics.

Page 2: Please change “…that were previously treated with and without antibiotics during DCT” to “…that were previously treated with and without DCT”.

Page 2: Please give information on the distribution of the number of cows in each farm and each group.

Page 2: Delete the word “throughout lactation” in the sentence “Colostrum and subsequent milk samples taken throughout lactation….”.

Page 3: Please move the drug information to their first indication on the objective.

Page 3: How many cows were excluded due to “Cows showing signs of mastitis or any other infections were excluded from the study and milk samples were collected only from healthy cows.”?

Page 3: Please describe the word “Bord Bia …. “.

Page 3: Please indicate the duration and the method that the samples were immediately frozen to get -20oC.

Page 4: Check English for the sub-heading “Microbiota diversity is influenced by antibiotic treatment”.

Page 4: Please describe Alpha, beta-diversity, Shannon and Chao1 diversity indices, and Bray-Curtis distance dissimilarity matrix in the Materials and Methods explaining also their relationship with the diversity of microbiota.

Page 4: Numbers of microbiota based on the metagenome result among groups should be described here.

Page 4 and 5: At 3.1: The explanation about the result contained many redundant words making it difficult to understand. Please check and reduce the unnecessary information. Figure 1 shows only Shannon, so, it is better to explain only explain Shannon, not Chao1. In addition, it is not necessary to explain all things because the readers can see the figure.

Page 5, 6: Please explain more about the benefits of the Bray-Curtis distance matrix in both text and figure to make the reader understand this.

Page 5: Please separate the effect of farm and parity into another paragraph.

Page 6: Check English (3.2) “Taxonomic composition is associated to dry cow therapy treatment”.

Page 6: Please indicate the number of a figure for this sentence “Overall, the phylum Actinobacteria was observed to have high relative abundance in both colostrum and milk samples in all groups with highest relative abundance in the NOAB group which was seen to decrease over time.”

Page 6-7: Figure 2A did not obviously show the composition of Bacteroidetes, please check.

Page 7: Why these references: (Por-cellato et al., 2021; Rubiola et al., 2020; McHugh et al., 2020; Bonsaglia et al., 2017; Zhang et al., 2015; Oikonomou et al., 2014; Gulbe & Valdovska 2014; Masoud et al., 2012) are in the result part.

Page 9: Please indicate the benefit of the songbird differential from the others.

Page 9: The sentence “Both previous and current antibiotic use may result in the selection of bacteria which carry resistance genes, thus increasing the overall resistome profile diversity of the microbiota.” Should be in the discussion part.

Page 10: Figure 4 should indicate the significance among groups

Reviewer 4 Report

The manuscript is well written and I have no further questions. However, what needs to be considered is the fact that the milk samples were taken over too long a time span of two months. I don't know if the microbiome and resisome in the sixth month of lactation has anything to do with the drying treatment? This is too long a period for the state of the microbiome to be attributed to drying therapy alone. Please explain this limitation. Today it is very fashionable to prove that cows pollute the environment with methane and that antibiotics are harmful. The strengthening of legislation in the last sentence of the Conclusion should be completely deleted, because even though it is mainstream thinking, the recommendation to the legislator is not part of the scientific concept, nor did you deal with legal aspects and regulations in this paper.

Author Response

Reviewer 4:

The manuscript is well written and I have no further questions. However, what needs to be considered is the fact that the milk samples were taken over too long a time span of two months. I don't know if the microbiome and resisome in the sixth month of lactation has anything to do with the drying treatment? This is too long a period for the state of the microbiome to be attributed to drying therapy alone. Please explain this limitation. Today it is very fashionable to prove that cows pollute the environment with methane and that antibiotics are harmful. The strengthening of legislation in the last sentence of the Conclusion should be completely deleted, because even though it is mainstream thinking, the recommendation to the legislator is not part of the scientific concept, nor did you deal with legal aspects and regulations in this paper.

Response 4:

We thank the reviewer for the comments and appreciate their constructive criticism and concerns, with a view to improving the quality of the current manuscript. We have now modified the manuscript by deleting unwanted conclusive remarks such as “and encourage the new European legislation discouraging use of blanket DCT” and “…support to the legislation that recommend” throughout the manuscript. We have also added and explained the limitation of larger sampling interval as follows “Another limitation of the study was the long sample intervals, which spans various seasonal and feeding variations, however all cows belonged to the same geographic location thus the effect of these variables, if any, was similar throughout the dataset”.

Round 2

Reviewer 3 Report

ARG is related to the routine use of antibiotics on each farm. This study was performed by separating the groups based on the farm shown on Page 3 “In terms of the farm-wise distribution of the cows included in our study, all NOAB cows were from a single farm: Farm C (n = 9), while farm D and P constituted the CEF group and had n = 4 and n = 4 respectively. Farm L and T formed the UBRO group and had n = 5 and n = 2 respectively.”. The authors need to inform the routine antibiotic use among the farms including dry cow therapy programs among the farms. Unless otherwise, the diversity of microbiota and resistome shown in this manuscript was due to the farm effect relating to the routine use of antibiotics.

Please also add the DCT policy of the farm whether blanket or selective DCT.

If the result of the study is related to the farm effect instead of the single intramammary infusion of antibiotics during the dry period, the change of the title might be a good option.

Please add the sentence in M&M about your experimental design in the “2.2 Sample and data collection”, for example, “Samples of colostrum (M0) and lactating milk from Months 2 (M2), 4 (M4), and 6 (M6) were collected for….”

3.1. Microbiota diversity is influenced by antibiotic treatment

Please explain the uses of both Shannon and Chao1. It is redundant and confusing. It would be better to select one index, or not? If so, you may say just “P<0.05”.

After giving the abbreviation, you do not need to use the full name again. Please delete all explained full names after the first use.

I am confused on the phase “(For UBRO: Shannon p-value=0.026, Chao1 p-value=0.011, For CEF: Shannon p-value=0.00016, Chao1 p-value=0.003; Wilcoxon test). Why do you have the word “Wilcoxon test” at the end?

I suggest the reduction of redundant and the full name of all paragraphs in the result, for example, changing the paragraph  of “3.1. Microbiota diversity is influenced by antibiotic treatment” toShotgun sequencing of the raw bovine colostrum and milk samples yielded 494,107,934 reads. Post trimming and host removal, the number of microbial reads remaining were 108,774,030 with an average of 1,121,382 reads (median = 310,854, min = 40,371, max = 6,941,451). Both Shannon and Chao1 diversity indices did not show variation between time points within the no-antibiotic group (Figure 1A, Figure S1A). However, significant differences were observed for both Shannon and Chao1 diversity indices between M0 and M2, M0 and M4, and M0 and M6) in both CEF and UBRO groups (Figure 1A, Figure S1A). Additionally, a significant difference between M2 and M6 was observed for the Chao1 index in the UBRO group (Figure S1A). Concerning individual time points, no significant differences were observed among groups at M0, while CEF and NOAB groups showed differences at M2; UBRO and NOAB at M4, and both antibiotics groups to NOAB at M6 (Figure 1B, Figure S1B). Beta diversity of milk microbiota between the NOAB and antibiotic groups (UBRO and CEF) demonstrated distinct clustering overall and at all time points after M0 (colostrum) (Figure 1C, 1D) including NOAB vs CEF (P=0.0030) and NOAB vs UBRO (P=0.0045) at M2, NOAB vs CEF (P=0.0015) and NOAB vs UBRO (p=0.0015) at M4, and NOAB vs CEF (P=0.0015) and NOAB vs UBRO at M6 (P=0.0015). Furthermore, discrete clustering of samples was observed based on time points, with samples at M0 clustering separately from all later time points (Figure 1D).

As the use of farm clustering for assigning the group, it would be difficult to analyze group and farm separately. The non-significant might be come from the small sample size in each farm compare to the sample size of the group. I would suggest deleting the following paragraph.

Similar to the distinct clustering of the NOAB group from antibiotic-treated groups overall, we also observed clustering based on farms, where farm “C” (NOAB) clusters separately from the other antibiotic-treated farms (farm “D” and “P” are CEF, while farm “L” and “T” are UBRO). Furthermore, samples from nulliparous cows which corresponded to NOAB group showed a similar clustering pattern because of overlapping between antibiotics groups and parity (Figure S2 A, B & C, Table S2). Based on PERMANOVA tests, the variable farms and antibiotics were the most explanatory variables regarding beta-diversity (Bray-Curtis distance) in our dataset (PER-MANOVA R2=0.15 and R2=0.11, reciprocally). To test the interactions between Antibiotics and Farm, we performed a PERMANOVA on the two most explanatory grouping variables. Results show that grouping based on antibiotic exposure explains higher variance compared to farms (R2 =0.11 and 0.03 and p-values=0.001 and 0.02 for Group and farm study variables respectively, PERMANOVA using Bray-Curtis distance ~ Antibiotic groups + farm).

3.2. Taxonomic composition is associated to dry cow therapy treatment

In the result, please describe the finding of your study. Any information related to other studies should state in “Discussion”. Delete the information related to previous studies.

Please revise this part to remove redundant words and sentences. For example:

In the NOAB group, the genera Actinoallotecihus, Corynebacterium, Brachybacterium, and Microbac-terium were amongst the top 10 over all time points (Figure 2). Brachybacterium and Brevibacterium showed an increase in NOAB compared with CEF and UBRO. Further, Corynebacterium was higher abundance at M2 in the NOAB group. Actinoalloteichus in all groups were high relative abundance at M0 and showed a declining trend with time of lactation (Figure 2C). Higher abundance of Acinetobacter was observed in the antibiotic-administered groups compared to NOAB (Figure 2B). Additionally, Rhodococcus, Ottowia, and Lactobacillus were observed in higher relative abundance in both the antibiotic-treated groups and demonstrated an increasing trend over time of lactation (Figure 2C). Pseudomonas, Mycobacterium, Corynebacterium, and Kocuria were also present at all time points in all three groups, with Kocuria and Microbacterium increasing over lactation time in the antibiotic-treated groups (Figure 2C).

A closer look at the relative abundance of major mastitis-causing pathogens in dairy cows revealed lack of high relative abundance of major mastitis pathogens such as Escherichia and Staphylococcus in the NOAB at M0 and M2 (Figure 3A). Relative abundance of Streptococcus was close to zero in all three groups (plot not shown). The relative abundance of Corynebacterium was higher in the NOAB group than the CEF group at M0, and higher than both antibiotic-treated groups at M2 (Figure 3A).

Delete the sentence “We also examined the effect of farm variable on the abundance of mastitis-causing pathogens and observed no significant differences (plot not shown).”.

I don’t see any differences between 3.2 and 3.3, please delete “3.3. Differential microbial composition in antibiotic vs non-antibiotic groups” and their description.

The description of Figure 3B did not appear in the text, please delete it.

3.4. Impact of dry cow therapy on antibiotic resistance gene reservoir

Please revise the paragraph after Figure 4 as the description in “3.1”.
